# Characteristics of self-motion sensation after major earthquakes: An internet survey

Reiko Tsunoda[1,2☯*], Tomohiko Kamo[1,3☯], Yumi Dobashi[1☯], Hiroaki Fushiki[1,2☯]

1 Mejiro University Ear Institute Clinic, Saitama, Japan, 2 Department of Speech, Language and Hearing Therapy, Faculty of Health Sciences, Mejiro University, Saitama, Japan, 3 Department of Physical Therapy, Faculty of Rehabilitation, Gunma Paz University, Takasaki, Gunma, Japan

☯ These authors contributed equally to this work.
* r.tsunoda@mejiro.ac.jp

## Abstract

This study aimed to determine the characteristics and prognosis of new-onset swaying or dizziness, defined as self-motion sensation, following the 2024 Noto Peninsula earthquake in Japan and identify factors associated with persistent symptoms and disturbance in daily life. A cross-sectional, internet-based survey was conducted 2 months after the earthquake among 1,000 residents of the affected regions. The incidence and characteristics of self-motion sensation, its impact on daily life, and symptom duration were assessed in the survey. Logistic regression analyses were performed to identify demographic factors (age, sex, and region), individual characteristics (dizziness treatment history, proneness to motion sickness, and trait anxiety), and emotional responses (feelings of anxiety or fear induced by self-motion sensation) associated with life disturbance and persistent symptoms. Of 968 valid responses, 43% of participants reported a new-onset self-motion sensation following the earthquake. Symptoms were described as brief episodes of body swaying lasting less than 1 min, mostly occurring while sitting or standing still. Among those with self-motion sensation, 18.5% reported disturbance in daily life, and 24.5% continued to experience symptoms 2 months after the event. Younger age and emotional responses were significantly associated with life disturbance. Proximity to the epicenter and emotional response were associated with persistent symptoms. Self-motion sensation is common after major earthquakes and usually resolves spontaneously. However, a subset of individuals experiences persistent symptoms or disturbances in daily life. Emotional responses, particularly anxiety or fear triggered by self-motion sensation, play crucial roles in daily life impact and symptom persistence. Early recognition and targeted support for at-risk individuals may help prevent long-term impairment and improve post-disaster health outcomes.

**Data availability statement:** All relevant data are within the paper and its Supporting Information files.

**Funding:** The author(s) received no specific funding for this work.

**Competing interests:** The authors have declared that no competing interests exist.

## Introduction

After major earthquakes, hospital visits for dizziness usually increase significantly [1,2], with additional reports indicating elevated inner ear or vestibular dysfunction [3,4]. Additionally, dizziness accompanied by headaches, along with insomnia and heightened nervousness, has been described as a common neuropsychiatric symptom following such events [5,6].

In Japan, the colloquial term jishin-yoi ("earthquake drunk") became common after the Great East Japan Earthquake on March 11, 2011. By April of that year, it was frequently referenced in otolaryngology clinical reports. In 2014, Nomura et al. formally defined this phenomenon as Post-Earthquake Dizziness Syndrome (PEDS) [7]. Subsequent studies, including a population-based survey following the 2016 Kumamoto earthquake in Japan [8] and multicenter investigations after the 2023 Kahramanmaraş earthquakes in Türkiye [9–13], have further documented PEDS in various seismic settings.

Many individuals have reported brief episodes of dizziness occurring in the absence of actual aftershocks [7,8]. However, it remains unclear whether this phenomenon constitutes a pathological condition. Moreover, the symptoms reported varied across different studies [2,7,8,14], and the prognosis and necessity of medical intervention remain partially elucidated.

On January 1, 2024, a magnitude 7.6 earthquake struck the Noto Peninsula in Ishikawa Prefecture, Japan. Systematic surveys are essential to evaluate the health impacts on affected populations and develop appropriate response measures. However, post-disaster studies should be conducted ethically and with careful consideration of timing to minimize the burden on survivors and avoid disrupting recovery efforts. Thus, we conducted an internet-based survey 2 months after the earthquake.

This study aimed to identify the characteristics of new-onset swaying or dizziness —defined here as self-motion sensation (SMS) — experienced after a major earthquake. Similarly, we sought to determine the factors associated with persistent symptoms or disturbance in daily life. Identifying these factors may inform early intervention strategies and support services for individuals at risk of persistent symptoms or functional impairment.

## Materials and methods

### Study design and participants

A cross-sectional internet-based survey was conducted between February 29 and March 1, 2024, 2 months after the Noto Peninsula earthquake. The target population comprised adults aged ≥18 years in Ishikawa, Toyama, and Fukui prefectures, including areas affected by the magnitude 7.6 earthquake that occurred on January 1, 2024 (Fig 1). The Japan Meteorological Agency reports earthquake shaking on a seismic intensity scale ranging from 0 (imperceptible) to 7 (extremely severe). During the Noto earthquake, seismic intensities ranged from 4 to 7 in Ishikawa, 4–5 in Toyama, and 3–5 in Fukui.

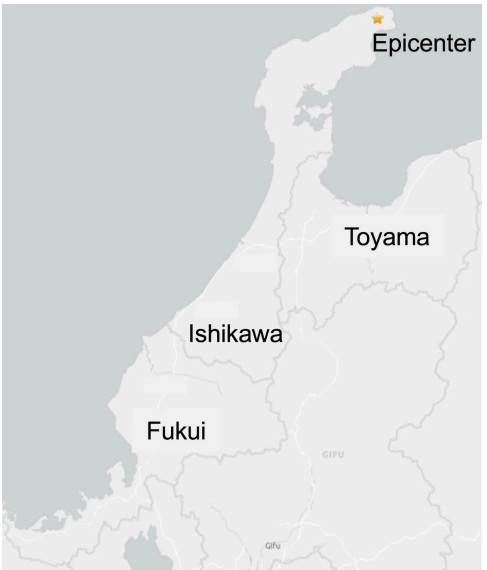

**Fig 1. Survey area and epicenter of the Noto Peninsula earthquake on January 1, 2024.** Information was reprinted from the United States Geological Survey (https://earthquake.usgs.gov/earthquakes/eventpage/us6000m0xl/map).

The survey was conducted online by Rakuten Insight, Inc. (Tokyo, Japan), which maintains a panel of 2.2 million registered monitors. To ensure internal consistency across participants, the response window was limited to 24 hours, and 1,000 responses were obtained. The questionnaire was used to collect information on age, sex, geographic location during the earthquake, new-onset swaying or dizziness, concomitant symptoms, individual characteristics (e.g., history of dizziness, motion sickness, and trait anxiety), the impact of symptoms on daily life, emotional responses (e.g., anxiety or fear), and symptom duration (S1 and S2 Files). The questionnaire was developed based on a previous study [8].

The primary outcomes were the characteristics of new-onset swaying or dizziness experienced after the earthquake: impact on daily life and symptom prognosis. The secondary outcomes included individual characteristics associated with disturbance in daily life and symptom persistence.

The study was approved by the Medical Research Ethics Committee of Mejiro University (approval number: Medical 23–055). The study purpose, survey methodology, information to be provided to the researcher, and publication method were clearly stated in writing on the survey website. Participants provided informed consent by checking a consent box on the study website.

## Statistical analysis

Logistic regression analyses were performed to calculate odds ratios (ORs) for associations between swaying or dizziness and clinical or demographic variables. Participants who reported that their daily life was "somewhat" or "much" disturbed were classified as experiencing disturbance in daily life. Those who reported ongoing symptoms 2 months after the earthquake were considered to have persistent symptoms.

To adjust for multiple testing across the two outcomes, Bonferroni correction was applied by multiplying P-values by 2, and statistical significance was defined as a Bonferroni-adjusted P-value <0.05. Sensitivity analysis using E-values was conducted to assess the potential influence of unmeasured confounding factors [15]. Analyses were performed using SPSS software, version 25.0 (IBM Corp., Armonk, NY, USA). The anonymized raw data used for this analysis are provided in S3 File.

## Results

### Incidence and characteristics of the swaying or dizziness following a major earthquake

A total of 1,000 responses were collected. After excluding 29 respondents who were not in the survey area during the main earthquake and three who did not indicate their sex, we analyzed data from 968 participants. The sample comprised 607 males and 361 females, with a mean age of 52.3 years (standard deviation, 12.38 years). In the survey area, the proportion of residents aged ≥70 years was high, and the data did not follow a normal distribution. The median age of the participants was significantly lower than that of the residents in the target area (median test, P = 0.000, Table 1). The Index of Dissimilarity between the sample and the target population was 31.4%.

After the January 1 earthquake, 416 of the 968 participants reported a swaying or dizziness sensation in the absence of actual seismic activity. They confirmed this as a novel sensation. The incidence rates, stratified using seismic intensity and region, are shown in Fig 2.

These sensations were primarily characterized as body swaying (87.8%), most of which lasted less than one minute (82.7%) (Fig 3A and 3B). The sensations occurred while participants were in stationary positions, such as sitting, standing, or lying down, and were not associated with body or head movements (Fig 3A). Half of the participants who experienced SMS reported feelings of anxiety or fear in response to the sensation (Fig 3C).

**Table 1. Age distribution.**

| Age | 18-19 | 20s | 30s | 40s | 50s | 60s | 70s | 80s- | Median (25, 75 percentile) |
|---|---|---|---|---|---|---|---|---|---|
| Participants | 0.2 | 4.0 | 11.9 | 23.5 | 29.4 | 24.0 | 6.4 | 0.6 | 53 (44, 62) |
| Target area | 2.3 | 10.4 | 11.2 | 15.3 | 16.4 | 14.5 | 16.9 | 13.0 | 56 (40.9, 72.6) |

Percentage of each age group in the population aged ≥18 years.

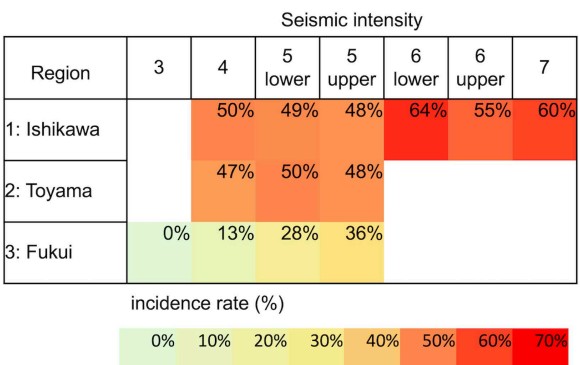

**Fig 2. Incidence rate of the swaying or dizziness sensation in the absence of an earthquake, based on seismic intensity and region.** The Japan Meteorological Agency announced the shaking of an earthquake as the seismic intensity. Seismic intensities ranged from 0 to 7. An earthquake with an intensity of 4 is considered "startling to most people, felt by most people walking, and most people are awoken." At Intensity 5 lower, "many people are frightened and feel the need to hold onto something stable," while at Intensity 6 upper, "it is impossible to remain standing or move without crawling; people may be thrown through the air." The Noto earthquake on January 1 had intensities of 4–7, 4–5, and 3–5 in Ishikawa, Toyama, and Fukui Prefectures, respectively. In Ishikawa Prefecture, over 50% of participants reported experiencing dizziness or a swaying sensation in the absence of an earthquake, noting it was their first time. In Toyama Prefecture, the proportion was approximately 50%, whereas in Fukui Prefecture, despite having a seismic intensity of 5 upper, 36% of participants reported similar sensations.

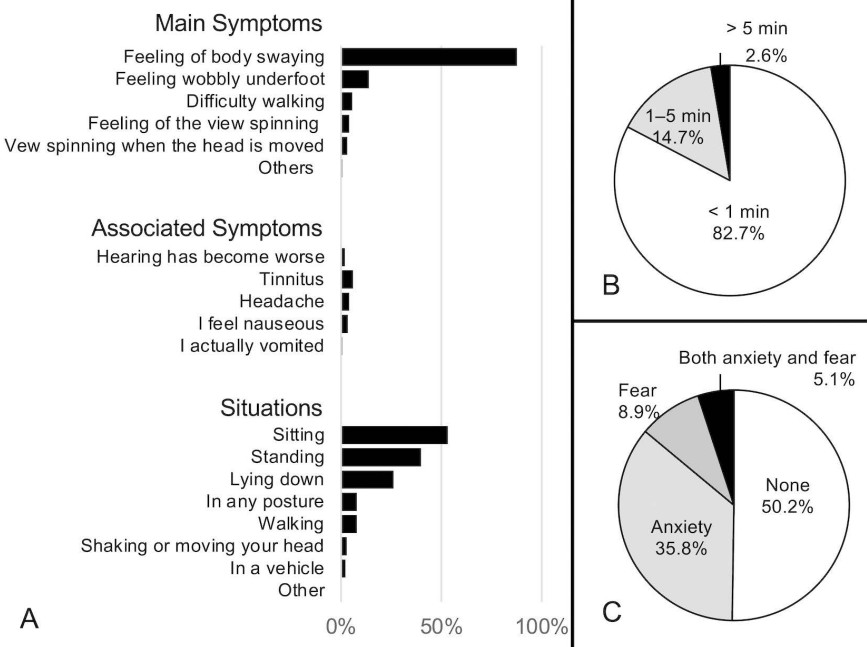

**Fig 3. Symptoms of swaying and dizziness in 416 participants who felt the sensation for the first time.** (A) This was mostly a self-motion sensation (SMS), characterized by body swaying, without associated symptoms. This was not spinning vertigo. SMS was felt while sitting, standing, lying down, or being still and was rarely accompanied by body or head movements. (B) Duration of SMS: symptom duration was mostly <1 min (82.7%) and rarely >5 min. (C) Feelings of anxiety or fear induced by SMS: Half of the participants with SMS reported anxiety or fear induced by the sensation.

## Impact on daily life and symptom persistence

Among participants who experienced SMS, 18.5% reported that their daily lives were "somewhat" or "much" disturbed owing to SMS (Fig 4).

Logistic regression analysis indicated younger age (OR: 0.588; 95% confidence interval (CI: 0.403–0.857; E-value = 1.93) and feelings of anxiety or fear induced by SMS (OR: 12.176; 95% CI: 5.574–26.598; E-value = 6.44) as significant factors associated with daily life disturbance (Table 2).

SMS resolved within 2 weeks in 61.3% of participants. However, 24.5% reported persistent symptoms 2 months after the earthquake (Fig 5).

Logistic regression analysis showed that proximity to the epicenter (OR: 0.638; 95% CI: 0.436–0.933; E-value = 1.81) and feelings of anxiety or fear induced by SMS (OR: 3.668; 95% CI: 2.202–6.110; E-value = 3.24) were significantly associated with symptom persistence (Table 3).

## Discussion

In this study, we conducted an internet-based survey targeting 1,000 residents in the disaster-affected areas 2 months after the 2024 Noto Peninsula earthquake. We focused specifically on newly developed swaying or dizziness symptoms, referred to as SMS, and excluded participants who reported only a worsening of pre-existing dizziness. Our findings revealed that 43% of participants experienced such sensations. Nomura et al. reported that 87.5% of individuals in Tokyo — approximately 300 km from the epicenter of the 2011 Great East Japan Earthquake — experienced brief, rocking sensations within 1 min, which they termed PEDS [7]. Similarly, Miwa et al. reported a 42.2% prevalence of post-earthquake

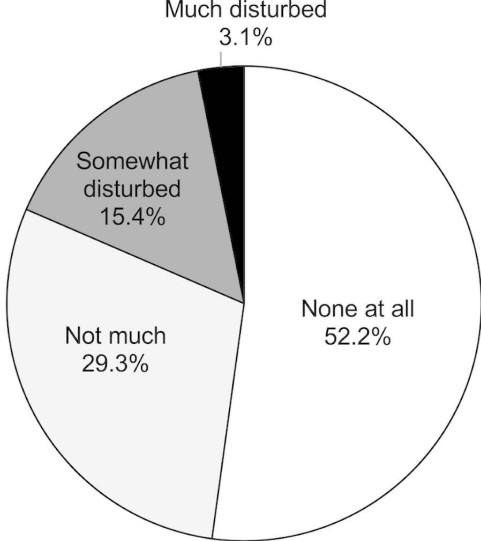

**Fig 4. Degree of disturbance in daily life owing to SMS.** SMS disturbed the daily lives of 18.5% of the participants.

**Table 2. Logistic regression analysis of factors associated with disturbance in daily life owing to SMS.**

| | Adjusted | | |
| --- | --- | --- | --- |
| | OR | 95% CI | P-value |
| Region | 0.792 | 0.516–1.218 | 0.576 |
| Sex | 1.495 | 0.809–2.764 | 0.400 |
| Age group | 0.588 | 0.403–0.857 | 0.012* |
| History of treatment for dizziness | 2.751 | 1.131–6.691 | 0.052 |
| Prone to motion sickness and/or seasickness | 1.458 | 0.778–2.731 | 0.480 |
| Trait anxiety | 1.436 | 0.751–2.746 | 0.546 |
| Feelings of anxiety or fear induced by SMS | 12.176 | 5.574–26.598 | 0.000** |

Coding of independent variables:

regions (1 = Ishikawa, 2 = Toyama, 3 = Fukui), sex (1 = male, 0 = female), age group (0 = ≤35 years, 1 = 36–50 years, 2 = 51–65 years, 3 ≥ 66 years), and other factors (1 = yes, 0 = no).

* P < 0.05

** P < 0.01

SMS, self-motion sensation; OR, odds ratio; CI, Confidence interval.

dizziness following the 2016 Kumamoto earthquake [8]. Although the methods and timeframes differ, these results consistently demonstrate a high prevalence of earthquake-related dizziness.

While previous studies referred to this condition as PEDS [7,14] or post-earthquake dizziness [8], we used the term SMS to describe the subjective experience of swaying or motion in the absence of seismic activity. SMS involved brief episodes of body sway lasting less than one minute, typically occurring while sitting or standing still, independent of head movement, and distinct from rotational vertigo. According to the Bárány Society [16], "dizziness" refers to disturbed spatial orientation without a false perception of motion, making it a less precise term for describing SMS. Notably, most symptoms resolved within 2 weeks, suggesting that SMS is not inherently pathological. In contrast, some studies have reported more persistent and complex symptoms following earthquakes. For example, Kumar et al. documented cases of

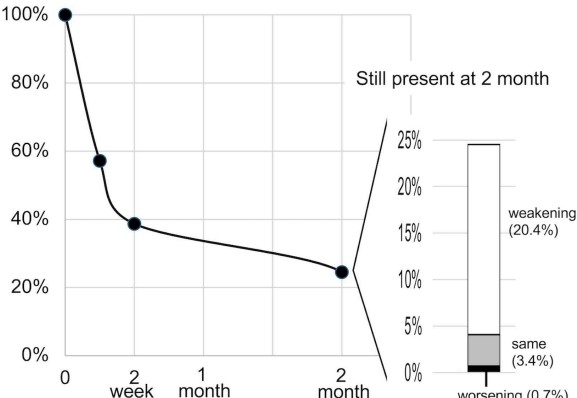

**Fig 5. Symptom persistence.** Over 60% of patients had SMS resolved within 2 weeks, whereas symptoms persisted in 24.5% 2 months after the earthquake.

**Table 3. Logistic regression analysis of factors associated with symptom persistence.**

|  | Adjusted | | |
|---|---|---|---|
|  | OR | 95% CI | P-value |
| Region | 0.638 | 0.436–0.933 | 0.042* |
| Sex | 0.979 | 0.578–1.656 | 1.0 |
| Age group | 1.278 | 0.930–1.755 | 0.260 |
| History of treatment for dizziness | 1.417 | 0.622–3.227 | 0.812 |
| Prone to motion sickness and/or seasickness | 0.930 | 0.525–1.648 | 1.0 |
| Trait anxiety | 1.940 | 1.076–3.498 | 0.056 |
| Feelings of anxiety or fear induced by SMS | 3.668 | 2.202–6.110 | 0.000** |

Coding of independent variables:

regions (1 = Ishikawa, 2 = Toyama, 3 = Fukui), sex (1 = male, 0 = female), age group (0 = ≤35 years, 1 = 36–50 years, 2 = 51–65 years, 3 = ≥66 years), and other factors (1 = yes, 0 = no).

* P < 0.05

** P < 0.01

SMS, self-motion sensation; OR, odds ratio; CI, Confidence interval.

continuous ground shaking, sustained fear, and imbalance. Symptoms included lightheadedness, swinging sensations, headaches, blackouts, and loss of consciousness [2]. Miwa et al. described PEDS as encompassing dizziness, vertigo, imbalance, nausea, and a persistent motion sensation, typically triggered soon after the earthquake [14]. These findings suggest that symptoms evolve over time and that SMS progresses into conditions that meet diagnostic criteria for vestibular or neuropsychiatric disorders. As our study was conducted 2 months after the earthquake and involved a tightly controlled 24-hour response window, it enabled a clear cross-sectional characterization of SMS during a relatively stable recovery phase.

Approximately 20% of participants with SMS reported disturbance in daily life, and 25% experienced persistent symptoms 2 months post-disaster. Persistent or impairing SMS may warrant clinical evaluation, and identifying affected individuals is crucial to timely intervention. Our analysis showed that younger age and emotional responses, such as feelings of anxiety or fear induced by SMS, were significantly associated with disturbance in daily functioning. Additionally, symptom persistence was associated with proximity to the epicenter and emotional responses. In our study, "feelings of anxiety or fear induced by SMS" reflects a form of state anxiety specifically related to earthquake-triggered symptoms. These findings are consistent with those

of previous studies. Kumar et al. reported frequent comorbid anxiety, panic, agoraphobia, and psychosomatic traits in patients with post-earthquake balance disorders [2]. Eker et al. found significantly elevated anxiety and depression scores among earthquake-affected individuals, as measured using the Hospital Anxiety and Depression Scale [9]. Homma et al. reported that high state anxiety scores, as measured using the State-Trait Anxiety Inventory, were significantly associated with equilibrium dysfunction in individuals exposed to aftershocks [17]. Çelebi et al., in a long-term follow-up conducted 1 year after the earthquake, demonstrated that trait anxiety, rather than state anxiety, better predicted persistent symptoms [11]. While our findings are consistent with these observations, they should be interpreted with caution. The cross-sectional design limits causal inference, and we did not use a standardized anxiety scale. Future studies should consider validated instruments such as the Hospital Anxiety and Depression Scale or the State-Trait Anxiety Inventory to better evaluate psychological factors.

The underlying mechanisms of SMS and the progression of its symptoms remain speculative. However, neural adaptation processes and vestibular system responses under extreme conditions, such as earthquakes, may play a role.

**Sensory reweighting:** The human vestibular system integrates visual, proprioceptive, and vestibular inputs. Following strong vestibular stimulation, such as that caused by an earthquake, heightened sensory vigilance may activate a fight-or-flight response. While sensory reweighting facilitates adaptation, it could be maladaptive, causing persistent or exaggerated perceptions of motion even in the absence of actual movement [14,18]. The questionnaire asked, "have you felt swaying or dizziness even though there was no actual earthquake at that time?" It was assumed that participants distinguished between aftershocks and SMS by checking for swaying of surrounding objects and earthquake information on the Internet. However, in hypersensitive individuals, even minimal aftershocks may be perceived as significant.

**Prediction errors and mismatch processing:** Earthquakes generate discrepancies between expected and actual sensory input, disrupting the brain's internal models of motion and spatial orientation. These prediction errors may contribute to prolonged motion sensations or postural instability [19].

**Psychophysiological interplay:** Anxiety and hypervigilance may amplify somatic sensations and impair vestibular compensation, sustaining SMS perception [20].

**Central sensitization and cortical plasticity:** Repetitive or intense vestibular input during earthquakes may cause maladaptive neuroplastic changes in the cortical regions responsible for spatial orientation. Functional imaging studies have implicated the temporoparietal junction and insular cortex in persistent dizziness and spatial disorientation [21,22].

Peripheral vestibular dysfunction may also play a role. Strong shaking during earthquakes may cause inertial stress on otoliths in the utricle, causing otolith dysfunction. Repeated shaking could result in otolith detachment, contributing to vestibular impairment or benign paroxysmal positional vertigo [1,3]. Eker et al. reported dysfunction in the left posterior semicircular canal among patients experiencing post-earthquake dizziness [9].

A vital question is whether SMS may progress into chronic vestibular conditions such as Persistent Postural-Perceptual Dizziness (PPPD). Factors associated with SMS-related daily life disturbances overlap with PPPD risk factors. Although PPPD worsens with upright posture and motion, whereas SMS usually occurs during rest, both involve altered vestibular perception. Longitudinal studies are required to clarify potential transitions. Early identification of individuals at risk may enable timely preventive interventions. Such studies should include ethically designed follow-up systems, and participants experiencing worsening symptoms should be referred to appropriate in-person psychological care.

Face-to-face or mail-based surveys were not feasible in the acute post-disaster phase owing to infrastructure damage and ethical constraints. Therefore, an internet-based approach was adopted for several reasons stated below:

**Minimizing psychological burden:** Individuals in the early aftermath of a disaster are usually coping with trauma, loss, and severe disruption. Internet surveys enable voluntary, anonymous participation at the respondent's pace and in their preferred setting.

**Avoiding interference with recovery efforts:** On-site surveys conducted in evacuation shelters or disaster areas may disrupt relief operations. Internet-based surveys eliminate the need for physical presence and reduce this risk.

**Ensuring autonomy and ethical compliance:** Participants can receive detailed information about study aims, data handling, and confidentiality, and consent can be obtained via a digital opt-in procedure.

**Enhancing accessibility and equity:** Internet-based surveys enable rapid, wide-reaching data collection across dispersed populations.

However, this strength is also a source of potential selection bias, a major limitation affecting the generalizability of Internet survey results [23]. Older adults, in particular, may be underrepresented owing to lower levels of digital literacy [24]. Our sample was biased toward younger individuals compared with the general population of the affected area; consequently, the representativeness of our sample was assessed using the Index of Dissimilarity, which yielded a value of 31.4%, indicating a moderate discrepancy between the sample and the target population. Specifically, individuals aged ≥70 years were underrepresented, while those aged 50–69 years were overrepresented. While previous studies have shown that post-earthquake dizziness is more common in younger to middle-aged adults [7,8], future studies should incorporate older individuals. Mixed-method approaches, such as caregiver-assisted surveys in shelters or nursing homes, may help mitigate this bias. However, they could introduce third-party influence [25].

Furthermore, because the survey was conducted 2 months after the earthquake, and most SMS symptoms resolve within 2 weeks, recall bias is a potential concern. Similarly, age-related differences in recall processes may have influenced symptom reporting. Older adults tend to underreport negative experiences in retrospective surveys, whereas younger adults are more likely to recall such experiences with increased intensity, particularly over longer recall periods [26,27]. Such age-related recall biases may explain some of the differences in disturbance of daily life owing to SMS between age groups observed in this study.

Additionally, self-selection bias should be considered, as participants may be drawn to the study based on personal interest or incentives [23].

Despite these limitations, the internet-based approach facilitated rapid data collection, minimizing participant burden. The survey was conducted 2 months after the earthquake, during a transitional period from the acute to the recovery phase, when many individuals had resumed daily routines [28]. Although recall bias is possible, this timing was considered ethically appropriate and accurate reporting was possible in cases of persistent symptoms. The 24-hour response window further ensured internal consistency and provided a high-resolution cross-sectional snapshot.

This study underscores the significance of addressing anxiety and fear elicited by SMS, which is a form of state anxiety specifically associated with earthquake-related symptoms. Accordingly, post-disaster surveys should be designed as research instruments and tools that provide direct, meaningful benefits to participants. Such surveys should aim to collect data and serve as gateways to support systems by offering accurate health information, screening for psychological symptoms, and facilitating access to mental health services. Achieving this goal requires coordinated efforts among researchers, clinicians, and local authorities [29]. Remote consultation systems involving professionals outside the disaster-affected area may further help alleviate the burden on local healthcare infrastructure while maintaining continuous support for those affected.

This study has some limitations. First, the cross-sectional design and lack of a non-exposed control group precluded causal inference. Although associations between SMS and factors such as anxiety or proximity to the epicenter were identified, these should be interpreted with caution. Second, the 2-month gap between the earthquake and the survey may have introduced recall bias, particularly for participants whose symptoms resolved quickly. Third, selection bias was possible owing to the underrepresentation of older adults. Finally, the long-term trajectory of SMS remains unclear. Future longitudinal studies with repeated assessments are required to clarify whether SMS represents an early stage of chronic dizziness conditions, such as PPPD.

## Conclusions

SMS after a major earthquake resolves spontaneously and may be a non-pathological response in most people. However, in this study, which was conducted 2 months after the 2024 Noto Peninsula earthquake, although a causal relationship

cannot be inferred owing to the cross-sectional study design, state anxiety was significantly associated with persistent symptoms and disturbance in daily life. Our findings highlight the significance of early identification and support for individuals at risk of symptom persistence and the importance of longitudinal studies in clarifying the potential progression of SMS to chronic vestibular disorders such as PPPD. Future post-disaster health strategies should incorporate psychological screening and support systems to mitigate long-term impacts and improve recovery outcomes.

## Supporting information

**S1 File. Questionnaire in Japanese.** Original language.
(PDF)

**S2 File. Questionnaire in English.**
(DOCX)

**S3 File. Dataset.** Anonymized raw data used for statistical analysis.
(XLSX)

## Acknowledgments

The authors thank M. Kawamura and R. Arai for their assistance in the preparation of this study. The authors thank Rakuten Insight, Inc. (Tokyo, Japan) for the data collection. The authors also thank Editage (www.editage.com) for the English language editing.

## Author contributions

**Conceptualization:** Reiko Tsunoda, Hiroaki Fushiki.

**Data curation:** Reiko Tsunoda.

**Formal analysis:** Tomohiko Kamo.

**Methodology:** Reiko Tsunoda, Yumi Dobashi.

**Supervision:** Hiroaki Fushiki.

**Visualization:** Reiko Tsunoda.

**Writing – original draft:** Reiko Tsunoda.

**Writing – review & editing:** Reiko Tsunoda, Tomohiko Kamo, Yumi Dobashi, Hiroaki Fushiki.

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
