## [Decision Letter · Decision Letter 0]

17 Apr 2025

Dear Dr. Tsunoda,

**I would like you to go carefully through all the reviews and comply with their recommendations, especially in the parts suggested by Reviewers 1 and 2:**

**1. how they suggest changing the sentences to make them more understandable**

**2. what they require you to change and explain in the methodology and presentation of results**

3. their suggestions how to make your paper suitable for PLOS ONE, especially in redefining terms that you have been using and how to connect them with existing terminology in this specific research area

**It is very important to respond to Reviewer 3 and explain what changes have been made according to the suggestions of the other reviewers.**

**I would like to see your paper published in PLOS ONE, but you must adhere to the journal's criteria and improve the paper to give it greater clarity, readability and justification through methods, results and improved discussion. Also, the references need changes (new, more suitable papers) to be in line with the reviewers' suggestions.**

We look forward to receiving your revised manuscript.

Kind regards,

Iskra Alexandra Nola

Academic Editor

PLOS ONE

**Journal Requirements:**

1. When submitting your revision, we need you to address these additional requirements. Please ensure that your manuscript meets PLOS ONE's style requirements, including those for file naming. The PLOS ONE style templates can be found at https://journals.plos.org/plosone/s/file?id=wjVg/PLOSOne_formatting_sample_main_body.pdf and https://journals.plos.org/plosone/s/file?id=ba62/PLOSOne_formatting_sample_title_authors_affiliations.pdf 2. We note that this data set consists of interview transcripts. Can you please confirm that all participants gave consent for interview transcript to be published? If they DID provide consent for these transcripts to be published, please also confirm that the transcripts do not contain any potentially identifying information (or let us know if the participants consented to having their personal details published and made publicly available). We consider the following details to be identifying information:- Names, nicknames, and initials- Age more specific than round numbers- GPS coordinates, physical addresses, IP addresses, email addresses- Information in small sample sizes (e.g. 40 students from X class in X year at X university)- Specific dates (e.g. visit dates, interview dates)- ID numbers Or, if the participants DID NOT provide consent for these transcripts to be published:- Provide a de-identified version of the data or excerpts of interview responses- Provide information regarding how these transcripts can be accessed by researchers who meet the criteria for access to confidential data, including:a) the grounds for restrictionb) the name of the ethics committee, Institutional Review Board, or third-party organization that is imposing sharing restrictions on the datac) a non-author, institutional point of contact that is able to field data access queries, in the interest of maintaining long-term data accessibility.d) Any relevant data set names, URLs, DOIs, etc. that an independent researcher would need in order to request your minimal data set. For further information on sharing data that contains sensitive participant information, please see: https://journals.plos.org/plosone/s/data-availability#loc-human-research-participant-data-and-other-sensitive-data If there are ethical, legal, or third-party restrictions upon your dataset, you must provide all of the following details (https://journals.plos.org/plosone/s/data-availability#loc-acceptable-data-access-restrictions):a) A complete description of the datasetb) The nature of the restrictions upon the data (ethical, legal, or owned by a third party) and the reasoning behind themc) The full name of the body imposing the restrictions upon your dataset (ethics committee, institution, data access committee, etc)d) If the data are owned by a third party, confirmation of whether the authors received any special privileges in accessing the data that other researchers would not havee) Direct, non-author contact information (preferably email) for the body imposing the restrictions upon the data, to which data access requests can be sent 3. We note that Figure 1 in your submission contain map images which may be copyrighted. All PLOS content is published under the Creative Commons Attribution License (CC BY 4.0), which means that the manuscript, images, and Supporting Information files will be freely available online, and any third party is permitted to access, download, copy, distribute, and use these materials in any way, even commercially, with proper attribution. For these reasons, we cannot publish previously copyrighted maps or satellite images created using proprietary data, such as Google software (Google Maps, Street View, and Earth). For more information, see our copyright guidelines: http://journals.plos.org/plosone/s/licenses-and-copyright. We require you to either present written permission from the copyright holder to publish these figures specifically under the CC BY 4.0 license, or remove the figures from your submission: a. You may seek permission from the original copyright holder of Figure 1 to publish the content specifically under the CC BY 4.0 license.   We recommend that you contact the original copyright holder with the Content Permission Form (http://journals.plos.org/plosone/s/file?id=7c09/content-permission-form.pdf) and the following text:“I request permission for the open-access journal PLOS ONE to publish XXX under the Creative Commons Attribution License (CCAL) CC BY 4.0 (http://creativecommons.org/licenses/by/4.0/). Please be aware that this license allows unrestricted use and distribution, even commercially, by third parties. Please reply and provide explicit written permission to publish XXX under a CC BY license and complete the attached form.” Please upload the completed Content Permission Form or other proof of granted permissions as an "Other" file with your submission. In the figure caption of the copyrighted figure, please include the following text: “Reprinted from [ref] under a CC BY license, with permission from [name of publisher], original copyright [original copyright year].” b. If you are unable to obtain permission from the original copyright holder to publish these figures under the CC BY 4.0 license or if the copyright holder’s requirements are incompatible with the CC BY 4.0 license, please either i) remove the figure or ii) supply a replacement figure that complies with the CC BY 4.0 license. Please check copyright information on all replacement figures and update the figure caption with source information. If applicable, please specify in the figure caption text when a figure is similar but not identical to the original image and is therefore for illustrative purposes only.The following resources for replacing copyrighted map figures may be helpful: USGS National Map Viewer (public domain): http://viewer.nationalmap.gov/viewer/The Gateway to Astronaut Photography of Earth (public domain): http://eol.jsc.nasa.gov/sseop/clickmap/Maps at the CIA (public domain): https://www.cia.gov/library/publications/the-world-factbook/index.html and https://www.cia.gov/library/publications/cia-maps-publications/index.htmlNASA Earth Observatory (public domain): http://earthobservatory.nasa.gov/Landsat:
http://landsat.visibleearth.nasa.gov/USGS EROS (Earth Resources Observatory and Science (EROS) Center) (public domain): http://eros.usgs.gov/#Natural Earth (public domain): http://www.naturalearthdata.com/

Reviewers' comments:

Reviewer's Responses to Questions

**Comments to the Author**

1. Is the manuscript technically sound, and do the data support the conclusions?

Reviewer #1: Yes

Reviewer #2: Yes

Reviewer #3: No

2. Has the statistical analysis been performed appropriately and rigorously?

Reviewer #1: Yes

Reviewer #2: Yes

Reviewer #3: No

3. Have the authors made all data underlying the findings in their manuscript fully available?

Reviewer #1: Yes

Reviewer #2: Yes

Reviewer #3: Yes

4. Is the manuscript presented in an intelligible fashion and written in standard English?

Reviewer #1: Yes

Reviewer #2: Yes

Reviewer #3: No

**Reviewer #1: ** Overall Assessment:

This study investigates the prevalence, characteristics, and prognostic factors of self-motion sensation (SMS) following the 2024 Noto Peninsula earthquake using an Internet survey. The topic is timely and relevant, particularly for disaster-affected regions. The manuscript is well-structured, and the methodology is appropriate for the research question. However, several issues need addressing before publication.

Major Comments:

1. Sample Representativeness and Bias:

o The participants were younger than the general population in the affected area, which may introduce selection bias. The authors acknowledge this limitation but should discuss its potential impact on the generalizability of the findings more thoroughly.

o The Internet survey method may exclude older or less tech-savvy individuals. Suggest including a comparison with demographic data from the affected regions to better contextualize the sample.

2. Terminology and Definitions:

o The term "self-motion sensation (SMS)" is introduced to distinguish the phenomenon from dizziness or vertigo. While the rationale is clear, the authors should justify this choice further by referencing additional literature or clinical guidelines.

o Clarify whether SMS is a novel classification or aligns with existing vestibular disorder frameworks (e.g., Persistent Postural-Perceptual Dizziness, PPPD).

3. Longitudinal Data and Chronicity:

o The study is cross-sectional, with data collected at a single time point (two months post-earthquake). A follow-up study or analysis of chronic cases (e.g., symptoms persisting beyond three months) would strengthen the conclusions about prognosis.

o Consider discussing the potential progression of SMS to chronic vestibular disorders, such as PPPD, and its implications for clinical management.

4. Anxiety as a Confounding Factor:

o Anxiety is identified as a key factor linked to prolonged SMS and daily life disruption. However, the study does not measure anxiety levels quantitatively (e.g., using standardized scales). Recommend citing literature on anxiety-vestibular interactions or suggesting future studies incorporate psychological assessments.

5. Clinical Implications:

o The discussion of "non-spinning vertigo" as a potential pathological outcome is intriguing but speculative. Provide more evidence or references to support this classification.

o Include practical recommendations for healthcare providers in disaster settings (e.g., screening for anxiety in individuals reporting SMS).

Minor Comments:

1. Figures and Tables:

o Tables 1–3 are well-presented, but consider simplifying the categorization explanations (e.g., age groups) for clarity.

2. Language and Clarity:

o The manuscript is generally well-written, but some sentences are overly complex (e.g., the Abstract’s conclusion). Minor editing for conciseness is recommended.

o Define abbreviations (e.g., SMS) at first use in the Abstract and Introduction.

3. References:

o The literature review is comprehensive, but some citations (e.g., media reports) could be replaced with peer-reviewed studies where possible.

Recommendation:

Major Revision Required. The manuscript addresses an important gap in disaster-related health research but requires revisions to address the above concerns, particularly regarding sample bias, terminology, and clinical implications. With these revisions, the study would be a valuable contribution to PLOS ONE.

Additional Suggestions:

• Include a brief paragraph on the cultural context of "earthquake drunk" in Japan to aid international readers.

• Discuss the ethical considerations of Internet surveys in post-disaster settings in more depth.

Final Note: The authors are commended for their timely research and clear presentation. Addressing these points will enhance the manuscript’s impact and readiness for publication.

Reviewer Decision: Revise and Resubmit

**Reviewer #2: ** This manuscript presents a cross-sectional internet survey investigating self-motion sensations (SMS) following the 2024 Noto Peninsula earthquake in Japan. While the topic is relevant and timely, there are several significant concerns that should be addressed before publication.

Major Concerns

The internet survey methodology introduces significant selection bias, particularly evident in the younger demographic skew of respondents. While acknowledged, this limitation warrants more extensive discussion regarding its impact on the validity and generalizability of findings.

The absence of a control group makes it impossible to determine whether SMS was actually caused by the earthquake rather than other factors.

The 2-month delay between the earthquake and data collection introduces substantial recall bias, especially for symptoms that may have resolved quickly. This limitation is inadequately addressed in the current manuscript.

The cross-sectional design cannot establish causality between anxiety and SMS persistence/disruption, yet causal language is sometimes implied in the discussion.

The authors fail to distinguish between pre-existing anxiety versus earthquake-induced anxiety, which could significantly confound the results.

The study lacks sensitivity analyses to evaluate the robustness of the findings.

The operational definition of SMS requires refinement. The distinction between SMS and aftershocks perception is inadequately addressed.

Given that proximity to epicenter was identified as a risk factor for prolonged symptoms, it is crucial to explain how participants distinguished between SMS and actual aftershocks, which were common in this region.

There is no clear operational definition distinguishing "self-motion sensation" from "dizziness," despite the attempt to discuss this in lines 176-185. This discussion lacks necessary rigor.

The paper lacks sufficient discussion of the physiological mechanisms underlying SMS. A more robust theoretical framework connecting vestibular function, anxiety, and seismic experiences would strengthen the manuscript.

Multiple comparisons were conducted without appropriate correction methods, raising concerns about Type I error.

Minor Concerns

There is an obvious grammatical error in lines 161-162: "...excluded from the study. that the results show that 43% of those who experienced a major earthquake also experienced an SMS." The sentence is fragmented and needs restructuring.

Several sentences are excessively long and complex (e.g., lines 115-118), reducing readability. These should be divided into shorter, clearer statements.

Terminology inconsistency exists between "disrupting daily life" in the abstract and "disturbing daily life" in the main text.

Some important conclusions lack direct citation support, such as the discussion of SMS characteristics in lines 173-180.

Provide an in-depth discussion of neural adaptation mechanisms and vestibular system responses to extreme experiences.

Explore the implications of research findings for post-earthquake mental health service provision.

Include a scatter plot or heat map illustrating the relationship between earthquake intensity and SMS.

Develop a visual representation of the multivariate risk prediction model.

Create a timeline chart showing symptom evolution over time.

**Reviewer #3: ** The effect of a large earthquake on people’s psychology and their daily life is the main topic which needs to pay more attentions. In this paper, the authors cast an internet survey on self-motion sensation by a questionnaire after the large 2024 7.6 earthquake in Noto Peninsula. 1000 responses have been received finally. They do some statistical work on the data and get some valuable results.

However, there are some uncertainties in the work:

Firstly of all, for a statistical investigation, the number of the 1000 samples in this time is too small to get a convincing result. Secondly, for a research paper, the investigating method (or technique) and the aim (or to solve a problem) are important. However, we cannot find any relevant contents in this paper. So this paper seems like a survey report instead of a research paper. Not mention that all figures are too simple and with low resolution. So a large amount of samples and scientific information are needed to support to this work if it is to publish in the future.

**Do you want your identity to be public for this peer review?** For information about this choice, including consent withdrawal, please see our Privacy Policy

Reviewer #1: No

Reviewer #2: No

Reviewer #3: No

---

## [Author Response · Author response to Decision Letter 1]

31 May 2025

1. This manuscript has been revised to meet the PLOS ONE's style requirements.

2. Regarding the use of interview transcripts:

The participants provided consent for the publication of the interview transcripts. In order to prevent the inclusion of any potentially identifying information, age has been categorized, and the transcripts have been revised accordingly.

3. Regarding Figure 1:

The map image in Figure 1 has been replaced with a version sourced from the USGS, which is in the public domain.

---

## [Decision Letter · Decision Letter 1]

3 Jul 2025

PLOS ONE

Dear Dr. Tsunoda,

Thank you for submitting your manuscript to PLOS ONE. After careful consideration, we feel that it has merit but does not fully meet PLOS ONE’s publication criteria as it currently stands. Therefore, we invite you to submit a revised version of the manuscript that addresses the points raised during the review process.

I received the responses from reviewers, and I would like you to address a few of the remaining comments that one of the reviewers (Reviewer #2) addressed. 

Please go through the above mentioned comments and add/change/check anything that is suitable to be improved. 

We look forward to receiving your revised manuscript.

Kind regards,

Iskra Alexandra Nola

Academic Editor

PLOS ONE

Journal Requirements:

Reviewers' comments:

Reviewer's Responses to Questions

**Comments to the Author**

Reviewer #2: All comments have been addressed

Reviewer #3: All comments have been addressed

2. Is the manuscript technically sound, and do the data support the conclusions?

Reviewer #2: Yes

Reviewer #3: Yes

3. Has the statistical analysis been performed appropriately and rigorously?

Reviewer #2: Yes

Reviewer #3: Yes

4. Have the authors made all data underlying the findings in their manuscript fully available?

Reviewer #2: Yes

Reviewer #3: Yes

5. Is the manuscript presented in an intelligible fashion and written in standard English?

Reviewer #2: Yes

Reviewer #3: Yes

Reviewer #2: The authors have made substantial and thoughtful revisions in response to previous comments. The inclusion of additional analyses (e.g., sensitivity analysis using E-values and Bonferroni correction), expanded discussions on methodological limitations (e.g., selection bias, recall bias), and theoretical mechanisms (e.g., sensory reweighting, prediction errors) have significantly strengthened the manuscript. Visualizations such as the heatmap and timeline chart are also a valuable addition, improving the clarity and presentation of key findings. Overall, the manuscript has been meaningfully improved and is closer to publication readiness.

Below are a few minor issues that require further clarification or refinement to ensure the manuscript meets the highest standards for publication.

Specific Comments:

1. Selection Bias and Representativeness

Comment: While the authors have included Table 1 showing the age distribution of participants and an expanded discussion on selection bias, the analysis of how this bias may influence the generalizability of findings could be slightly more detailed.

Suggestion: Consider quantifying the representativeness of the sample (e.g., by comparing participant demographics to census data or using representativeness indices). This would provide a more concrete assessment of the impact of selection bias.

2. Recall Bias

Comment: The authors acknowledged recall bias due to the 2-month delay in data collection and justified this timing. However, the potential variability in symptom reporting due to recall bias remains underexplored.

Suggestion: Briefly discuss how recall bias may have influenced specific findings (e.g., over-reporting or under-reporting of SMS) and whether this bias may differ between younger and older participants. A sentence or two in the discussion would suffice.

3. Operational Definition of SMS

Comment: The revised definition of SMS is clearer and better distinguishes it from dizziness. However, the explanation of how participants distinguished SMS from aftershocks could be slightly expanded.

Suggestion: Consider adding a brief sentence clarifying whether participants were explicitly instructed to differentiate SMS from aftershocks (e.g., through self-checks or online earthquake reports).

4. Language and Clarity

Comment: While the authors have improved the manuscript's language, a few sentences remain overly complex or slightly ambiguous.

Reviewer #3: The authors have answered the questions and comments one by one and minior mistakes are corrected. The manuscript could be accepted as it is.

**Do you want your identity to be public for this peer review?** For information about this choice, including consent withdrawal, please see our Privacy Policy

Reviewer #2: No

Reviewer #3: No

---

## [Author Response · Author response to Decision Letter 2]

23 Jul 2025

Response to journal requirements:

We have removed the following reference, as only the abstract was available and we were unable to access the full text.

Tevzadze N, Shakarishvili R. Vertigo syndromes associated with earthquake in Georgia. Georgian Med News. 2007;148-149: 36-39.

Two additional references related to recall bias have been added.

26. Charles ST, Piazza JR, Mogle JA, Urban EJ, Sliwinski MJ, Almeida DM. Age Differences in Emotional Well-Being Vary by Temporal Recall. J Gerontol B Psychol Sci Soc Sci. 2015; 71:798–807. DOI: https://doi.org/10.1093/geronb/gbv011

27. Junghaenel DU, Broderick JE, Schneider S, Wen CKF, Mak HW, Goldstein S, et al. Explaining age differences in the memory-experience gap. Psychol Aging. 2021�36: 679–693. DOI: https://doi.org/10.1037/pag0000628

---

## [Editor Report · Decision Letter 2]

1 Aug 2025

Characteristics of self-motion sensation after major earthquakes: An Internet survey

PONE-D-25-05484R2

Dear Dr. Reiko Tsunoda,

We’re pleased to inform you that your manuscript has been judged scientifically suitable for publication and will be formally accepted for publication once it meets all outstanding technical requirements.

Kind regards,

Iskra Alexandra Nola

Academic Editor

PLOS ONE
---

## [Editor Report · Acceptance letter]

PONE-D-25-05484R2

PLOS ONE

Dear Dr. Tsunoda,

I'm pleased to inform you that your manuscript has been deemed suitable for publication in PLOS ONE. Congratulations! Your manuscript is now being handed over to our production team.

Kind regards,

on behalf of

Dr. Iskra Alexandra Nola

Academic Editor

PLOS ONE